

# Association of serum fetuin-B with insulin resistance and pre-diabetes in young Chinese women: evidence from a cross-sectional study and effect of liraglutide

Xuyun Xia[1], Shiyao Xue[2], Gangyi Yang[2], Yu Li[3], Hua Liu[4], Chen Chen[5] and Ling Li[1]

[1] The Key Laboratory of Laboratory Medical Diagnostics in the Ministry of Education and Department of Clinical Biochemistry, College of Laboratory Medicine, Chongqing Medical University, Chongqing, China
[2] Department of Endocrinology, The Second Affiliated Hospital, Chongqing Medical University, Chongqing, China
[3] Department of Pathology, Chongqing University Cancer Hospital, Chongqing, China
[4] Department of Pediatrics, University of Mississippi Medical Center, Jackson, MS, The United States of America
[5] Endocrinology, SBMS, Faculty of Medcine, University of Queensland, Brisbane, Australia

Corresponding author
Ling Li, liling@cqmu.edu.cn

## ABSTRACT

**Background and Aims.** Fetuin-B has been reported to be involved in glucose and lipid metabolism and associated with the occurrence of diabetes. The main purpose of this study is to explore the changes of circulating fetuin-B in young women with pre-diabetes and to analyze the relationship between fetuin-B and the occurrence and development of IR.

**Methods.** A total of 304 women were enrolled in this study and subjected to both OGTT and EHC. A subgroup of 26 overweight/obese womenwas treated with Lira for 24 weeks. serum fetuin-B concentrations were measured by ELISA.

**Results.** In IGT and IR-NG groups, serum fetuin-B levels were higher than those in the NGT group. The serum fetuin-B levels in the IGT group were higher than those in the IR-NG group. serum fetuin-B was positively correlated with BMI, WHR, 2h-BG, FIns, HbA1c, and HOMA2-IR, but negatively correlated with the M-value in all study populations. Multiple stepwise regression analysis showed that the M-value was independently and inversely associated with serum fetuin-B. Logistic regression analysis showed that serum fetuin-B was independently associated with IGT and significantly increased the risk of IGT. During the OGTT, serum fetuin-B increased significantly in the NGT group, but there were no significant changes in other groups. During the EHC, serum fetuin-B increased in the IGT group, but there was no change in other groups. After Lira intervention, serum fetuin-B decreased significantly in IGT women.

**Conclusions.** serum fetuin-B levels are elevated in young women with IR or IGT and may be associated with IR.

## INTRODUCTION

Type 2 diabetes mellitus (T2DM) is a metabolic disease characterized by relative insulin deficiency caused by impaired islet function and target organ insulin resistance (IR). As the main epidemic area of diabetes in Asia, China has become the country with the largest number of diabetic patients in the world (*Chatterjee, Khunti & Davies, 2017*). Traditionally, T2DM has been considered a chronic disease related to the elderly, but with rapid economic and social development in recent years, changes in lifestyle and diet structure have resulted in increasing rates of T2DM and obesity in adolescents (*Arcidiacono et al., 2020*; *Lascar et al., 2018*). According to a 2010 epidemiological survey in China, the rate of diabetes in China is 4.5% in adolescents. The rate of pre-diabetes is as high as 40–50% in the population under 40 years old (*Xu et al., 2013*). It has been reported that the pathogenesis of youth T2DM is mainly related to the rapid decline of islet β - cell function and obesity-induced IR. In young individuals with obesity and T2DM, the levels of circulating inflammatory factors, adipokines, hepatocytes, and myokines were significantly changed (*Lascar et al., 2018*; *Reinehr et al., 2016*), but the underlying mechanisms are unclear.

serum fetuin-B is considered to be an adipo-hepatokine and a member of the cystatin superfamily of cysteine protease inhibitors. It is mainly secreted by the liver, and a small amount is also secreted by white adipose tissue (WAT) and myocardial cells (*Olivier et al., 2000*). The results from *Meex et al. (2015)* showed that fetuin-B could induce IGT by inhibiting the effect of insulin on myotubes and hepatocytes in mice. In obese mice, knockout of the fetuin-B gene significantly improved impaired glucose tolerance (IGT). In an *in vivo* study, it was shown that fetuin-B may regulate blood glucose through a non-insulin signaling pathway, but the specific mechanism is still unclear (*Meex et al., 2015*). In a clinical study, Qu et al. found that the serum fetuin-B levels in T2DM patients were significantly higher than those in IGT and normal subjects, and serum fetuin-B was correlated with TG, fasting insulin, homeostasis model assessment of insulin resistance (HOMA-IR), and insulin secretion in the first phase (*Qu et al., 2018*). In addition, it has been reported that serum fetuin-B levels are significantly increased in patients with nonalcoholic fatty liver disease (NAFLD), and related to IR (*Li et al., 2018*). In contrast, a recent study showed that the mRNA expression of fetuin-B in the liver was not associated with HOMA-IR (*Peter et al., 2018*). Therefore, the relationship between serum fetuin-B and IR is still controversial.

In this study, the changes in serum fetuin-B were observed in simple IR, IGT, and normal subjects, and the relationship between serum fetuin-B and other metabolic indexes was evaluated. In addition, because off-label use of liraglutide (Lira) has been proposed for the management of PCOS in women with overweight/obesity-related IR, therefore, in a subgroup of overweight/obese women, the effects of Lira on fetuin-B and glucose metabolism were investigated.

# RESEARCH POPULATION AND METHODS

## Research population

A total of 304 young women were enrolled in this study (Fig. 1), including 80 newly diagnosed IR patients (IR-NG group), 97 individuals with impaired glucose tolerance (IGT group), and 127 healthy women (NGT group). The individuals were recruited from the outpatient department of Endocrinology or from daily physical examination, the Second Affiliated Hospital of Chongqing Medical University from January 2017 to December 2019. IGT was diagnosed according to the American Diabetes Association (ADA) (*American Diabetes, 2012*). The presence of pre-diabetes was defined by the occurrence of IGT at 2-hours oral glucose tolerance (2h-OGTT). M-value <6.28 was considered to have IR (*Wei-Gang et al., 2009*). Individuals with M-value <6.28 and normal fasting blood glucose (FBG) levels were included in the IR-NG group. All IGT and IR patients were newly diagnosed without any medication or lifestyle intervention. The exclusion criteria included type 1 diabetes mellitus (T1DM), T2DM, hypertension, and other important organ diseases such as heart, liver, and kidney. NGT group had normal blood glucose [FBG <5.1 mmol; 2-h blood glucose post-glucose load (2-h BG) <7.8 mmol], no family history of diabetes and hypertension, no chronic diseases, and medication recently. All subjects signed informed consent before the experiment. This study was conducted according to the declaration of Helsinki and approved by the ethics committee of the Second Affiliated Hospital of Chongqing Medical University (2014 Colombo Review No. (72)). Human research was registered at the Chinese Clinical Trial Registry (registered 23 June 2011), chictr.org.cn (CHICTR-OCC-11001422).

## Anthropometric and biochemical measurements

After fasting for 10–12 h, all subjects were given a physical examination and biochemical tests by professionals. The physical examination included body weight, blood pressure (BP), waist circumference (WC), and body fat content (Fat %). Biochemical parameters included total cholesterol (TC), triglyceride (TG), high-density lipoprotein cholesterol (HDL-C), low-density lipoprotein cholesterol (LDL-C), free fatty acid (FFA), HbA1c, FBG, 2-hour postprandial blood glucose (2h-BG), fasting insulin (FIns), and 2-hour postprandial insulin (2h- INS), as previously reported (*Xu et al., 2020*).

## Determination of serum fetuin-B concentration

Serum fetuin-B concentration was determined by an ELISA kit (RayBiotech, Inc. Norcross, GA, USA). The detection limit for fetuin-B was 4.0 ng/ml. Intra- and inter-assay variations (CV) were 10% and 12%, respectively. This method has high sensitivity, good specificity, and no obvious cross-reaction.

## OGTT and euglycemic-hyperinsulinemic clamp test (EHC)

After 10–12 h of fasting overnight, all subjects were given 75g glucose orally and the OGTT test was performed at 8:00 AM (*American Diabetes, 2012*). During the OGTT, blood samples were collected for the measurements of blood glucose, insulin, and serum fetuin-B at 0, 30, 60, and 120 min. The EHC was performed in all subjects. The operation process was
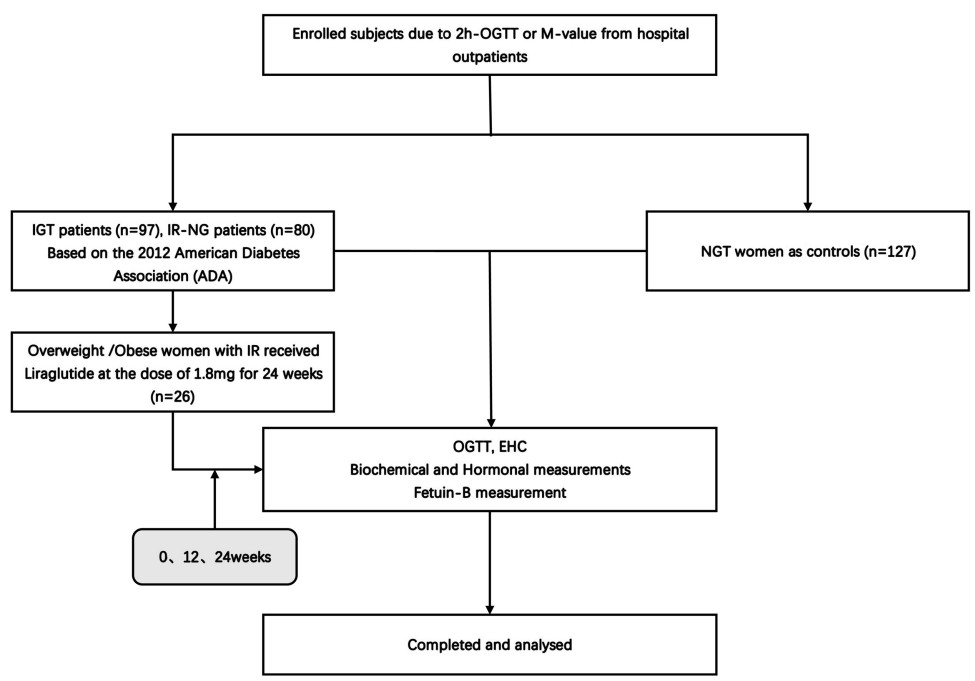

**Figure 1** Study flow diagram.

as previously reported (*Xu et al., 2020*). During the EHC, human insulin (1mU/kg/min) was continuously infused for 2 h. At the same time, the infusion rate of the 20% glucose was adjusted according to the blood glucose levels, tested every 15 min, to maintain blood glucose at the basic level (4.5–5.5 mmol/L). In the steady-state of the clamp, GIR was equal to glucose disposal rate (GDR), and M-value was related to GIR and body weight. Blood samples were collected at 0, 80, 100, and 120 min for the determination of fetuin-B and insulin. Blood samples were centrifuged, and serum was separated and stored at −80 °C for subsequent analysis.

## GLP-1RA intervention

A total of 26 overweight/obese women with IR participated in GLP-1RA intervention for 24 weeks. These women were enrolled for intervention with lira due to ovarian dysfunction and refractory menstrual problems. Liar, a GLP-1RA, was injected subcutaneously once a day. The dose increased from 0.6 mg to 1.8 mg. Inclusion criteria included age of 18–35 and BMI of 25–35 kg/m$^2$. Exclusion criteria included the family history of thyroid tumor, the history of medullary thyroid carcinoma, severe gastrointestinal disease, acute pancreatitis, and pregnancy. Those subjects did not use any drugs in the last 3 months. Physical examination and biochemical tests, OGTT, and EHC tests were performed before treatment, 12 and 24 weeks after treatment. All subjects signed informed consent before treatment.

## Calculation formula

BMI = body weight/height 2, WHR = WC/hip circumference (HC), HOMA2-IR was calculated by the software (HOMA calculator v2.2.2) (*Wallace, Levy & Matthews, 2004*), M-value = GIR/body weight, and M-value <6.28 is considered to have IR (*Wei-Gang et al., 2009*).

## Statistical analysis

SPSS version 24.0 software (SPSS 2.0, Chicago, IL) was used for statistical analysis. Data were expressed as mean ± SD or median. Variables that are not normally distributed were converted logarithmically. Analysis of variance was used for comparison among groups, and Fisher's Least Significant Difference (LSD) test was used for comparison between two groups. The relationship between serum fetuin-B and other variables was evaluated by linear regression analysis with controlled covariates. Multiple linear regression analysis was used to identify independent factors associated with serum fetuin-B. Multiple logistic regression analysis was conducted to understand the association of serum fetuin-B with IR and IGT. ROC curves were constructed to evaluate the sensitivity and specificity of serum fetuin-B in predicting IR and IGT.

We calculated tolerance and variance inflation factor (VIF) values to evaluate multicollinearity between variables. A tolerance <0.1 and VIF >10 is considered indicative of multicollinearity. Compared with the control group, $p < 0.05$ was considered to be statistically significant.

## RESULTS

### Serum fetuin-B levels and biochemical parameters in the study population

Table 1 shows the clinical biochemical indexes and circulating fetuin-B levels in each group. We found that BMI, Fat %, WC, BP, TG, TC, LDL, HbA1c, FBG, 2 h-BG, FIns, 2 h-INS, HOMA2-IR, the area under the curve of glucose (AUCg) and insulin (AUCi) in IR-NG and IGT groups were significantly higher than those in the control group (all $p < 0.01$), while HDL-C and M-values were lower ($p < 0.01$). In addition, FFA, 2h-BG, 2h-Ins, and AUCg in the IGT group were significantly higher than those in the IR-NG group ($p < 0.05$ or $p < 0.01$). serum fetuin-B levels in IGT and IR-NG groups were significantly higher than those in normal controls ($p < 0.05$ or $p < 0.01$, Table 1 and Fig. 2A). fetuin-B levels were higher in the IGT group than those in the IR-NG group (Fig. 2A, $p < 0.05$). Furthermore, serum fetuin-B levels in obese/overweight women (BMI $\geq$ 24 kg/m$^2$) were significantly higher than those in normal controls (7.63 ± 0.38 *vs.* 5.95 ± 0.32 mg/L, $p < 0.01$) (Fig. 2B). The relationship between serum fetuin-B and IR was analyzed by combing IR-NG and IGT individuals into the IR group. The results showed that serum fetuin-B levels were significantly correlated with IR (Table S2).

 

**Table 1  Anthropometrics and metabolic parameters in study population.**

| Variable | NGT ($n = 127$) | IR-NG ($n = 80$) | IGT ($n = 97$) |
|---|---|---|---|
| Age (years) | 26.5 ± 3.0 | 27.2 ± 4.5 | 27.5 ± 4.0 |
| BMI (kg/m$^2$) | 20.2 ± 2.2 | 26.3 ± 3.6[**] | 26.1 ± 4.2[**] |
| FAT (%) | 26.5 ± 5.1 | 38.0 ± 5.9[**] | 37.4 ± 6.6[**] |
| WC (cm) | 69.5 ± 7.0 | 86.1 ± 8.9[**] | 86.5 ± 9.7[**] |
| SBP (mmHg) | 107.9 ± 8.5 | 116.5 ± 11.4[**] | 116.4 ± 12.9[**] |
| DBP (mmHg) | 71.4 ± 9.1 | 74.7 ± 9.4[*] | 75.0 ± 9.5[**] |
| TC (mmol/L ) | 3.88 ± 0.90 | 4.40 ± 0.79[**] | 4.60 ± 1.02[**] |
| TG (mmol/L)[†] | 0.81 (0.60–1.13) | 1.48 (1.04–2.12)[**] | 1.59 (1.21–2.14)[**] |
| HDL-C (mmol/L)[†] | 1.24 (0.99–1.51) | 1.10 (0.94–1.28) | 1.17 (1.02–1.30)[**] |
| LDL-C (mmol/L) | 2.20 ± 0.76 | 2.60 ± 0.56[**] | 2.78 ± 0.86[**] |
| FFA (μmol/L)[†] | 0.51 (0.36–0.73) | 0.48 (0.37–0.65) | 0.56 (0.45–0.78)[**,Δ] |
| HbA1c (%)[†] | 5.10 (5.00–5.30) | 5.40 (5.20–5.60)[**] | 5.50 (5.30–5.80)[**] |
| FBG (mmol/L)[†] | 4.54 (4.26–4.91) | 5.26 (4.98–5.50)[**] | 5.35 (5.04–5.70)[**] |
| 0.5h-BG (mmol/L)[†] | 7.40 (6.49–8.45) | 9.06 (8.28–9.79)[**] | 9.64 (8.86–10.78)[**,Δ] |
| 1h-BG (mmol/L)[†] | 6.07 (4.99–7.20) | 8.14 (7.17–9.55)[**] | 10.10 (9.38–11.43)[▲] |
| 2h-BG (mmol/L)[†] | 5.42 (4.66–6.25) | 6.47 (5.97–7.20)[**] | 8.73 (8.10–9.40)[▲] |
| FIns (mU/L)[†] | 7.14 (5.94–8.73) | 18.42 (12.00–24.63)[**] | 20.39 (13.90 –27.95)[**] |
| 0.5h-Ins (mU/L)[†] | 81.96 (54.38–115.84) | 146.50 (96.98–250.40)[**] | 135.85 (83.48–198.80)[**] |
| 1h-Ins (mU/L)[†] | 54.64 (36.80–83.61) | 162.65 (115.70–251.48)[**] | 171.25(121.15 –241.35)[**] |
| 2h-Ins (mU/L)[†] | 41.57(24.21–63.11) | 118.70 (89.73–182.33)[**] | 197.15 (129.58–289.20)[▲] |
| AUCi [†] | 107.5 (61.1–139.0) | 277.0 (188.6–406.2)[**] | 293.1 (206.1–442.2)[**] |
| AUCg | 11.7 (10.3–13.4) | 15.0 (14.2–16.6)[**] | 18.4 (17.2–19.5)[▲] |
| HOMA2-IR[†] | 1.00 (0.84–1.27) | 2.77 (1.76–3.58)[**] | 2.99 (2.06–4.11)[**] |
| Fetuin-B (mg/L) | 5.38 ± 3.50 | 6.89 ± 3.61[*] | 8.13 ± 4.86[**,Δ] |
| M-value[†] | 10.29 (8.63–11.46) | 4.32 (3.59–5.13)[**] | 3.98 (2.98–5.29)[**] |

**Notes.**

Values are given as mean ± SD or median (Inter quartile Range).

BMI, body mass index; FAT (%), the percentage of fat in body; WC, waist circumference; SBP, systolic blood pressure; DBP, diastolic blood pressure; TG, triglyceride; TC, total cholesterol; HDL-C, high-density lipoprotein cholesterol; LDL-C, low-density loprotein cholesterol; FFA, free fatty acid; FBG, fasting blood glucose; 0.5h-BG, 0.5 h blood glucose after glucose overload; 1h-BG, 1-h blood glucose after glucose overload; 2h-BG, 2-h blood glucose after glucose overload; FIns, fasting plasma insulin; 2h-Ins, 2-h plasma insulin after glucose overload; AUCi, the area under the curve for insulin; AUCg, the area under the curve for glucose; HOMA-IR, homeostasis model assessment of insulin resistance.

[†]Log transformed before analysis

[*]$p < 0.05$.

[**]$p < 0.01$ compared with the NGT group.

[Δ]$p < 0.05$.

[▲]$p < 0.01$ compared with IR-N.

## Association of serum fetuin-B with clinical and biochemical indexes in young women

In all study subjects, serum fetuin-B levels were positively correlated with obesity and IR-related parameters (BMI, fat%, WC, TG, FIns, 2h-INS, AUCi, AUCg, HOMA2-IR, and M-value) (all $p < 0.01$), but not with TC, HDL-C, LDL-C and FFA (Table 2). After controlling age and Fat%, serum fetuin-B levels were still significantly positively correlated with WC, FIns, HOMA2-IR, and M-value ($p < 0.05$ or $p < 0.01$, Table 2). Multiple regression analysis

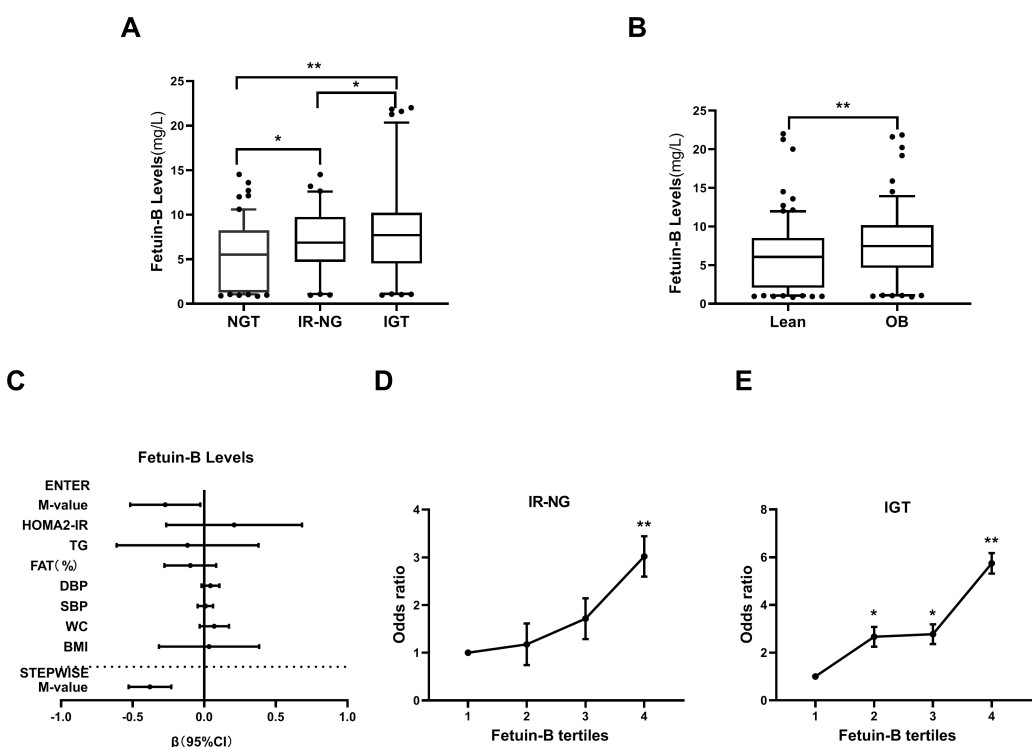

**Figure 2  Serum fetuin-B levels in the study population.** (A) Circulating fetuin-B levels in NGT, IR-NG and IGT subjects. (B) Circulating fetuin-B levels according to BMI (lean: BMI < 24 kg/m2; overweight/obese: BMI ≥ 24 kg/m2). (C) Circulating fetuin-B levels, according to M-values (IR: M-values < 6.28; non-IR: M-values ≥ 6.28). (D) All factors and stepwise multiple regression analyses of the serum fetuin-B in study individuals. (E) The odds ratio of having IR-NG in different quartiles of serum fetuin-B (quartile 1: 0.85−2.12 mg/L; quartile 2: 2.13−6.35 mg/L; quartile 3: 6.36−8.61 mg/L; quartile 4: >8.61 mg/L). (F) The odds ratio of having IGT in different quartiles of serum fetuin-B (quartile 1: 0.85−3.39 mg/L; quartile 2: 3.40−6.69 mg/L; quartile 3: 6.70−8.99 mg/L; quartile 4: >8.99 mg/L) Data were means ± SME. *$p < 0.05$ or **$p < 0.01$ *vs.* Controls, lean, no-IR or quartile 1.

showed that M-value was an independent factor affecting serum fetuin-B (Fig. 2C, Table 2). The regression equation is $Y_{fetuin-B} = 9.231 − 0.379X$ M-value.

## Association of serum fetuin-B with IR and IGT in women

By multiple logistic regression analysis, we found that serum fetuin-B levels were significantly correlated with the occurrence of IR-NG and IGT (IR-NG: OR, 1.13; 95% CI [1.04–1.22]; $p < 0.01$; IGT: OR, 1.20; 95% CI [1.09–1.27], $p < 0.01$). After control of age, Fat %, BMI, WC, BP, and TG, serum fetuin-B levels were still correlated with IGT ($p < 0.05$ or $p < 0.01$, Table S1), while by controlling age only, fetuin-B was correlated with the occurrence of IR in women ($p < 0.01$, Table S2).

To further analyze the association of serum fetuin-B with IR-NG and IGT, we divided serum fetuin-B concentrations into four quartiles in NGT and IR-NG groups (Quartile 1, 0.85−2.12 mg/L; Quartile 2, 2.13−6.35 mg/L; Quartile 3, 6.36−8.61 mg/L; Quartile 4, >8.61 mg/L), and in NGT and IGT groups (Quartile 1, 0.85−3.39 mg/L; Quartile 2, 3.40−6.69

**Table 2  Linear regression analysis of variables associated with serum fetuin-B levels in young woman.**

| Variable | Model 1 | | Model 2 | | Model 3 | |
|---|---|---|---|---|---|---|
|  | r | p | r | p | B | p |
| Age (years) | 0.042 | 0.473 | | | | |
| FAT (%) | 0.195 | < 0.01 | | | | |
| BMI (kg/m2) | 0.242 | < 0.001 | 0.119 | 0.057 | | |
| WC (cm) | 0.252 | < 0.001 | 0.191 | < 0.05 | | |
| SBP (mmHg) | 0.166 | < 0.01 | 0.077 | 0.222 | | |
| DBP (mmHg) | 0.146 | < 0.05 | 0.102 | 0.106 | | |
| TC (mmol/L ) | 0.089 | 0.127 | 0.015 | 0.807 | | |
| TG (mmol/L)[†] | 0.150 | < 0.05 | 0.056 | 0.374 | | |
| HDL-C (mmol/L)[†] | −0.049 | 0.397 | −0.029 | 0.650 | | |
| LDL-C (mmol/L) | 0.088 | 0.130 | 0.009 | 0.887 | | |
| FFA (μmol/L)[†] | −0.038 | 0.515 | −0.030 | 0.636 | | |
| HbA1c (%)[†] | 0.201 | < 0.001 | 0.125 | 0.046 | | |
| FBG (mmol/L)[†] | 0.171 | < 0.01 | 0.067 | 0.288 | | |
| 2h-BG (mmol/L)[†] | 0.219 | < 0.001 | 0.108 | 0.087 | | |
| FIns (mU/L)[†] | 0.261 | < 0.001 | 0.172 | < 0.05 | | |
| 2h-Ins (mU/L)[†] | 0.220 | < 0.001 | 0.095 | 0.132 | | |
| AUCi [†] | 0.182 | < 0.01 | 0.063 | 0.315 | | |
| AUCg | 0.155 | < 0.01 | 0.031 | 0.627 | | |
| HOMA2-IR[†] | 0.262 | < 0.001 | 0.170 | < 0.05 | | |
| M-value[†] | −0.299 | < 0.001 | −0.230 | < 0.001 | −0.379 | < 0.001 |

Notes.
Model 1 unadjusted simple linear regression analysis.
Model 2 adjusted age and FAT%, partial linear regression analysis.
Model 3 multiple linear stepwise regression analysis, values included for analysis were BMI, WC, SBP, DBP, FAT%, TG, HOMA2-IR and M-value.
[†] Log transformed before analysis.

mg/L; Quartile 3, 6.70−8.99 mg/L; Quartile 4, >8.99 mg/L). As shown in Figs. 2D and 2E, the risk of IGT and IR increased with the increase of serum fetuin-B concentration.

## ROC curve analysis

To evaluate the predictive value of serum fetuin-B levels for IGT and IR, ROC curves were drawn. The results showed that the area under the ROC curve for IGT (AUC $_{IGT}$) was 0.67, sensitivity 79%, and specificity 56%. The best cut-off value of fetuin-B for predicting IGT was 4.45 mg/L (Fig. S1A). AUC$_{IR}$ was 0.63, sensitivity 72%, and specificity 51%. The best cut-off value of fetuin-B for predicting IR was 5.44 mg/L (Fig. S1B).

## Effect of the OGTT and EHC on serum fetuin-B levels in women

To further explore the regulatory factors of serum fetuin-B level, we performed the OGTT and EHC experiments. During the OGTT, serum fetuin-B levels in the NGT group were significantly increased (Fig. 3A) and reached the peak at 30 min (11.42 ± 1.0 mg/L *vs.* 3.42 ± 0.57 mg/L, $p < 0.01$), but no change in both IR-NG and IGT groups. The areas under the curve of serum fetuin-B (AUC fetuin-B) in the IGT and IR-NG groups were significantly higher than those in the control group (Fig. 3B). During the EHC experiment, as expected,

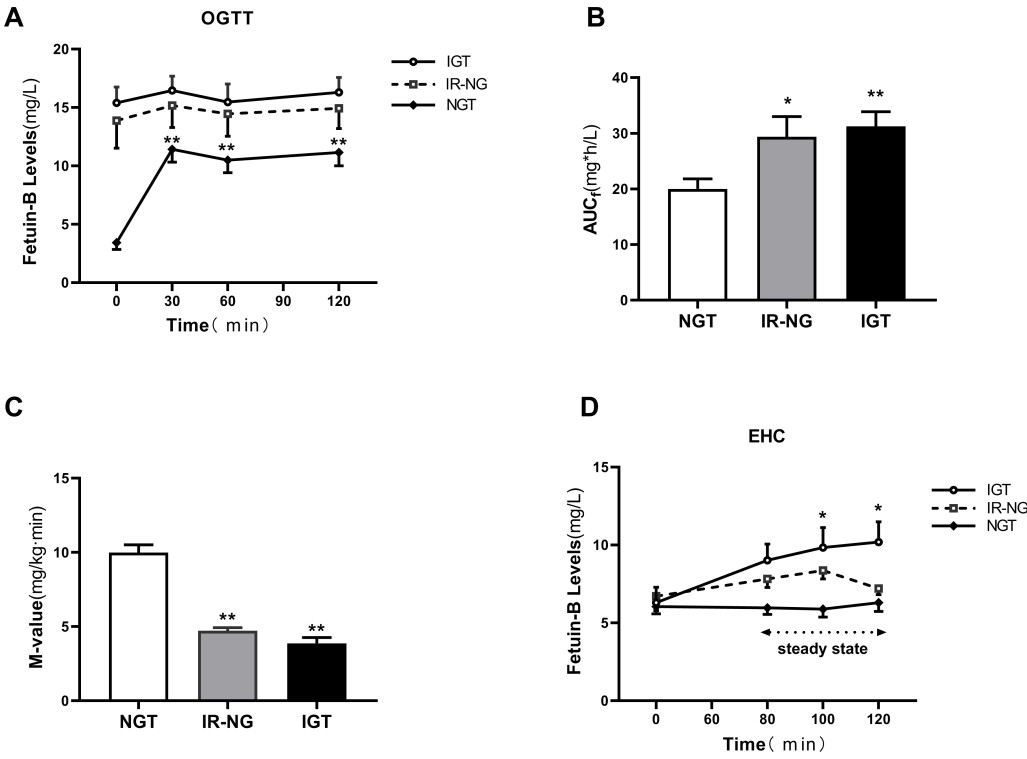

**Figure 3** **Circulating fetuin-B levels in interventional studies.** (A) Time course of changes in circulating fetuin-B levels in healthy, IR-NG and IGT subjects during the OGTT; (B) The area under the curve for serum Fetuin-B ($AUC_f$) during the OGTT. (C) The M-values in healthy, IR-NG and IGT subjects during the EHC. (D)Time course of circulating fetuin-B changes in healthy, IR-NG and IGT subjects during the EHC; data were meant ± SME. $^*p < 0.05$ or $^{**}p < 0.01$ *vs.* Control or baseline.

the M-values in IR-NG and IGT groups were significantly lower than those in the NGT group ($4.72 \pm 0.21$ and $3.86 \pm 0.39$ *vs.* $9.98 \pm 0.53$ mg/kg/min; $p < 0.01$; Fig. 3C). In the IGT group, serum fetuin-B levels increased gradually following the stimulation of hyperinsulinemia (from $6.29 \pm 0.99$ mg/L to $10.18 \pm 1.30$ mg/L) (Fig. 3D). Interestingly, there was no significant change in serum fetuin-B levels in both IR-NG and NGT groups during the EHC, suggesting that serum fetuin-B levels were regulated by insulin in different glucose tolerance populations.

## Effect of GLP-1RA treatment on serum fetuin-B level in obese individuals

Twenty-six overweight/obese women with IR (BMI, $28.4 \pm 0.6$ kg/m2) were treated with a Liar for 24 weeks. The changes in general clinical and biochemical indexes before and after treatment are shown in Table 3. After the treatment, obesity, and lipid metabolism-related indicators (BMI, Fat %, WC and TG), glucose metabolism and IR-related indicators (FBG, FIns, HbA1c, HOMA2-IR) were significantly improved compared with those before treatment ($p < 0.01$ or $p < 0.05$; Table 3). In addition, Liar treatment significantly decreased the M-values of the EHC in these obese women, which further indicates the improvement

**Table 3** Main clinical and metabolic features pre- and post-treatment with GLP-1RA in IGT and IR-NG women.

| Variable | Baseline | Post-treatment 3 months | Post-treatment 6 months |
|---|---|---|---|
| BMI (kg/m2) | $28.43 \pm 3.13$ | $26.44 \pm 2.97^{*}$ | $25.91 \pm 3.38^{**}$ |
| Fat (%) | $39.99 \pm 5.54$ | $36.25 \pm 4.12^{*}$ | $35.88 \pm 5.02^{**}$ |
| WC (cm) | $88.86 \pm 7.25$ | $85.98 \pm 7.10$ | $84.56 \pm 7.60^{*}$ |
| SBP (mmHg) | $116.72 \pm 11.72$ | $111.32 \pm 10.23$ | $112.68 \pm 12.33$ |
| DBP (mmHg) | $74.08 \pm 8.53$ | $73.80 \pm 10.08$ | $73.24 \pm 9.24$ |
| TC (mmol/L) | $4.74 \pm 0.74$ | $4.26 \pm 0.89^{*}$ | $4.33 \pm 0.82$ |
| TG (mmol/L) | $1.89 \pm 0.64$ | $1.55 \pm 0.67$ | $1.41 \pm 0.68^{*}$ |
| HDL-C (mmol/L) | $1.14 \pm 0.26$ | $1.05 \pm 0.18$ | $1.08 \pm 0.19$ |
| LDL-C (mmol/L) | $3.00 \pm 0.71$ | $2.63 \pm 0.82$ | $2.62 \pm 0.80$ |
| FFAs (μmol/L) | $0.50 \pm 0.17$ | $0.44 \pm 0.12$ | $0.45 \pm 0.22$ |
| HbA1c (%) | $5.50 \pm 0.36$ | $5.26 \pm 0.26^{**}$ | $5.20 \pm 0.24^{**}$ |
| FBG (mmol/L) | 5.37 (5.14–5.47) | 5.17 (5.03–5.46) | 5.14 (4.93–5.24)$^{*}$ |
| 0.5h-BG (mmol/L) | 9.28(8.60–10.23) | 8.66 (7.58–10.15) | 8.59 (7.67–9.42)$^{*}$ |
| 1h-BG (mmol/L) | 9.14(7.62–10.48) | 8.46 (6.74–9.85) | 8.38 (7.49–9.79) |
| 2h-BG (mmol/L) | 7.81(6.82–8.83) | 6.84 (5.59–7.97) | 6.71 (5.73–8.29)$^{*}$ |
| FIns (mU/L) | 21.69 (16.00–31.07) | 17.66 (11.43-24.13)$^{*}$ | 13.42 (9.71 -22.59)$^{**}$ |
| 0.5h-Ins (mU/L) | 172.95 (126.80–210.10) | 145.70 (93.27–240.38) | 142.05 (91.20-209.55) |
| 1h-Ins (mU/L) | 173.70 (119.30–334.10) | 179.35 (108.73–261.58) | 185.20 (121.95-257.08) |
| 2h-Ins (mU/L) | 174.70 (119.95–269.58) | 154.25 (78.09–252.93) | 121.05 (78.76-194.08) |
| AUCi | 285.3 (224.3–556.5) | 304.3 (176.4–466.0) | 290.8 (192.6-399.5) |
| AUCg | 16.9 (14.9–18.3) | 15.5 (13.0–17.3) | 15.0 (13.9-17.4) |
| HOMA2-IR | 0.83 (0.78–0.85) | 0.78 (0.76–0.85) | 0.77 (0.73-0.80)$^{*}$ |
| M-value | $3.76 \pm 0.73$ | $5.11 \pm 1.70^{**}$ | $5.27 \pm 1.79^{**}$ |
| Fetuin-B (mg/L) | $8.08 \pm 1.77$ | $7.47 \pm 1.48$ | $6.75 \pm 1.57^{*}$ |

**Notes.**

Values were given as mean $\pm$ SD or median (interquartile range).

$^{*}p < 0.05$.

$^{**}p < 0.01$ vs. Baseline.

$^{\Delta}p < 0.05$.

$^{\blacktriangle}p < 0.01$ vs. post-treatment 3 months.

of IR (Fig. 4A). Importantly, serum fetuin-B levels were significantly reduced after the 24 weeks of GLP-1RA intervention in these obese women with IR (from $8.08 \pm 1.77$ mg/l to $6.75 \pm 1.57$ mg/l, $p < 0.05$; Fig. 4B).

## DISCUSSION

In this study, the changes in serum fetuin-B concentrations in women with IGT or IR alone were reported for the first time. We found that (1) serum fetuin-B levels in women with IGT or IR alone were significantly higher than those in the normal control group and significantly correlated with glucose and lipid metabolism; (2) in these individuals, the M-value was an independent influencing factor for serum fetuin-B levels; (3) with the increase of serum fetuin-B levels, the risk of IR and IGT was also increased; (4) The response of serum fetuin-B to hyperglycemia and hyperinsulinemia was different in

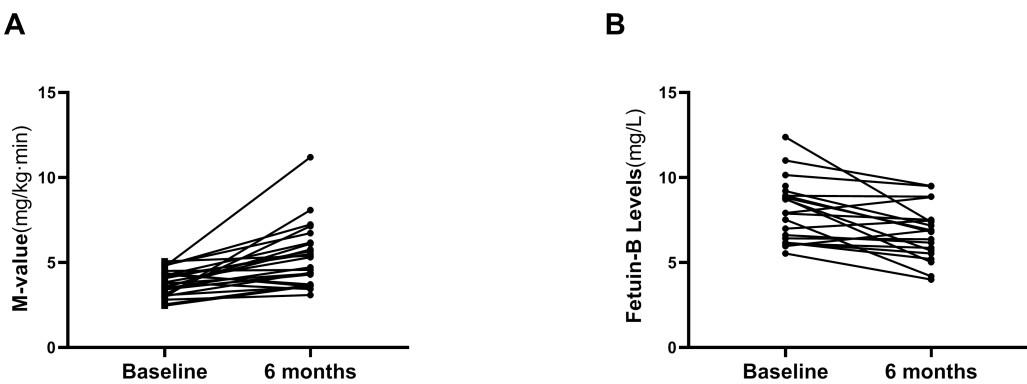

**Figure 4** **Effects of GLP-1RA treatment on serum fetuin-B in young women.** (A) Changes of M-value in IGT or IR subjects during the EHC after GLP-1RA treatment. (B) Serum fetuin-B levels in IGT or IR subjects after GLP-1RA treatment. Data were means ± SME. *$p < 0.05$ or **$p < 0.01$ *vs.* Baseline.

different glucose tolerance population; (5) GLP-1RA treatment not only improved IR but also reduced serum fetuin-B levels. These results suggest that fetuin-B may be a biomarker related to IR and glucose and lipid metabolism.

In recent years, there have been some reports that fetuin-B is related to metabolic diseases. *Meex et al. (2015)* found that serum fetuin-B was significantly increased in obese individuals with fatty liver disease and was associated with fasting insulin and HOMA-IR, but not with BMI and blood lipids. Other studies also found that serum fetuin-B levels were significantly increased in T2DM, NAFLD, and GDM patients, and were associated with IR and obesity (*Kralisch et al., 2017*; *Li et al., 2018*). In contrast, some studies found that serum fetuin-B levels were not related to BMI (*Qu et al., 2018*), and there was no significant correlation between serum fetuin-B and hepatocyte fat content in NAFLD patients (*Ebert et al., 2017*). Therefore, the results from previous studies have not been consistent. However, most of the previous studies were cross-sectional and descriptive studies. They were merely a preliminary exploration of fetuin-B, and there was no intervention experiment to fully prove the relationship between fetuin-B and human IR, and no state-of-the-art methodology was used. In addition, there have been many confounding factors in previous studies, such as middle-aged and elderly subjects, most of whom had long-term dyslipidemia or other metabolic diseases, and with a long history of drug treatment. In addition, age and gender also have important effects on obesity and IR. Therefore, if applied to clinical work, the previous results have some limitations.

In the present study, we further confirm that serum fetuin-B levels were significantly increased in women with IR and IGT, and significantly correlated with BMI, HOMA-IR, and glucose and lipid metabolism. This was consistent with the results of Adamska et al. in the PCOS population (*Adamska et al., 2019*). Based on the following reasons, we believe that our results are accurate and reliable: (1) we selected young subjects as the research objects and thus avoided age-related IR; (2) we chose women as the research subjects and avoided the influence of sex hormones on insulin sensitivity *in vivo*; (3) our subjects did

not use drug treatments or lifestyle interventions, avoiding interventional influence on serum fetuin-B levels.

In the current study, there is a contradictory finding that the cut-off value of fetuin-B for predicting IR was higher than that of predicting IGT. The reason for this phenomenon remains unknown. We speculate that serum fetuin-B levels may be mainly affected by insulin, and there was no hyperinsulinemia in a small number of the IGT individuals. In addition, IGT individuals with IR were also included in the calculation of the IR cut-off value.

To further explore the regulation of glucose and insulin on fetuin-B levels, we conducted an oral OGTT and an EHC. Under OGTT induced hyperglycemia and hyperinsulinemia, the serum fetuin-B levels in the control group increased significantly at different times, while the serum fetuin-B levels in IGT and IR-NG groups did not change significantly. We speculate that in individuals with IGT or IR, long-term IR results in the decline or failure of fetuin-B secretion function in tissues and cells. Therefore, there was no response to high glucose and high insulin stimulation. However, it is not known if hyperglycemia or/and hyperinsulinemia stimulates the secretion or release of fetuin-B *in vivo*. Therefore, we performed the EHC study, a gold standard for IR evaluation, and found that acute hyperinsulinemia resulted in significantly elevated circulating fetuin-B levels in IGT subjects under normal glucose conditions. However, acute hyperinsulinemia did not cause changes in circulating fetuin-B in IR-NG or control groups. We, therefore, believe that in the normal population, acute hyperglycemia increases the release and secretion of fetuin-B *in vivo*, while hyperinsulinemia does not stimulate the secretion of fetuin-B. However, the underlying mechanism needs further study.

Metformin has been widely reported for the treatment of obesity and polycystic ovary syndrome (PCOS) (*Gilbert, Valois & Koren, 2006*; *Cassina et al., 2014*). Although metformin can effectively reduce blood glucose, its effect of reducing body weight and improving IR is relatively limited (*Li et al., 2015*). Recently, GLP-1RA has been used for the treatment of obese individuals, and it has been observed that GLP-1RA is superior to metformin in improving IR, but there is still controversy on its effect on body weight reduction (*Mirabelli et al., 2019*; *Lin et al., 2020*). In addition, GLP-1 has been found to stimulate or inhibit adipokine releases, such as adiponectin and visfatin (*Li et al., 2011*; *Li et al., 2008*). To further observe the effect of GLP-1RA and insulin sensitivity on circulating fetuin-B, we treated overweight/obese patients with Lira, a GLP-1RA, for 24 weeks, and observed the changes in serum fetuin-B levels. After 24 weeks of Lira treatment, circulating fetuin-B levels in IGT individuals decreased significantly with weight loss and increased with insulin sensitivity. It has been well known that Lira is generally used at doses up to 1.8 mg as an anti-diabetic agent, but higher doses (up to 3.0 mg) have been approved for weight loss in the US and Europe in obese individuals, irrespective of hyperglycemia. Given the weight characteristics of the Chinese population, we treated IR patients with 1.8 mg Lira for 24 weeks. The results showed that body weight was also significantly reduced at this dose. Interestingly, menstrual cycles was not significantly changed in our study, maybe because of the small number of participants. We are recruiting a larger scale of patients in another trial to further study that. Based on these results, we speculate that the effect of

Lira on fetuin-B is likely a result of the weight loss and/or metabolic changes and improved IR after Lira treatment.

Our research has some limitations, including that (1) our study is a cross-sectional study, so we can not infer the causal relationship between fetuin-B and the occurrence and development of IR; (2) our study population is made up of only Chinese women; the findings may not be applicable to men and other ethnic populations. (3) A small sample size, especially in an intervention trial, may lead to biased results; (4) as a secretory protein, serum fetuin-B may be secreted by pulse, and the results of a blood sample may not fully reflect the true levels of circulating fetuin-B *in vivo*; (5) Lira treatment was a self-control experiment with no placebo control group. Nevertheless, we believe that our results are still sufficient to prove the association between fetuin-B and IR and guide other studies.

## CONCLUSIONS

Our results suggest that serum fetuin-B levels were elevated in young women with IGT or IR and might be associated with glucose metabolism and IR. In addition, the secretion and release of fetuin-B may be regulated by blood glucose. Therefore serum fetuin-B may be a biomarker for pre-diabetes and IR in young women.

## ACKNOWLEDGEMENTS

We thank patients and healthy individuals who made this study possible.

### Funding

This work was supported by research grants from the National Natural Science Foundation of China (No. 81670755 and 81873658). The funders had no role in study design, data collection and analysis, decision to publish, or preparation of the manuscript.

### Grant Disclosures

The following grant information was disclosed by the authors:
National Natural Science Foundation of China: 81670755, 81873658.

### Competing Interests

The authors declare there are no competing interests.

### Author Contributions

- Xuyun Xia and Shiyao Xue conceived and designed the experiments, performed the experiments, analyzed the data, prepared figures and/or tables, and approved the final draft.
- Gangyi Yang conceived and designed the experiments, prepared figures and/or tables, and approved the final draft.
- Yu Li analyzed the data, authored or reviewed drafts of the paper, reviewed and edited the manuscript, and approved the final draft.

- Hua Liu and Chen Chen conceived and designed the experiments, authored or reviewed drafts of the paper, reviewed and edited the manuscript, and approved the final draft.
- Ling Li conceived and designed the experiments, authored or reviewed drafts of the paper, and approved the final draft.

## Human Ethics

The following information was supplied relating to ethical approvals (i.e., approving body and any reference numbers):

This study was conducted according to the declaration of Helsinki and approved by the ethics committee of the Second Affiliated Hospital of Chongqing Medical University (2014 Colombo Review No. (72)).

Human research was prospectively approved and registered at the Chinese Clinical Trial Registry (registered 23 June 2011, http://www.chictr.org.cn) under registration number CHICTR-OCC-11001422.

## Data Availability

The raw measurements are available in the Supplementary File.

## Supplemental Information

Supplemental information for this article can be found online at http://dx.doi.org/10.7717/peerj.11869#supplemental-information.

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
