# Peer review of "Association of serum fetuin-B with insulin resistance and pre-diabetes in young Chinese women: evidence from a cross-sectional study and effect of liraglutide"

_PeerJ, doi:10.7717/peerj.11869_

## Round 0.1 · original submission · Major Revisions

The authors should address all the concerns raised by the reviewers.

Reviewer 1 ·

Basic reporting

In this cross-sectional study, with a preliminary 6-month interventional investigation, Xia and colleagues address the association of serum Fetuin B with Insulin Resistance (IR) and prediabetes in young Chinese women. Despite of an adequate sample size (127 healthy normal weight women, 80 insulin resistant but normal glucose tolerant women, 97 women with IR and prediabetes), and its interesting findings, unfortunately, the present article is overshadowed by several major flaws and complicated details, that make it difficult to fully appreciate the authors’ work.

-Title
“Serum Fetuin-B levels were elevated in young women with pre-diabetes”. There is no need of using past tense in the title, given that the effects of liraglutide intervention on Fetuin B represented a secondary analysis in a (very) small subgroup of patients. Something like “Association of serum Fetuin B with insulin resistance and pre-diabetes in young Chinese women: evidence from a cross-sectional study and effect of liraglutide” might be more appropriate.

-Abstract
Contrary to what has been stated, the aim of this cross-sectional study was not to analyze “the relationship of between the changes of Fetuin-B and the occurrence and development of IR” (line 28). For this goal, a large prospective observational study should have been designed. Lines 30-32 should be rephrased: “A total of 304 women were enrolled in this CROSS-SECTIONAL study AND SUBJECTED TO BOTH OGTT AND EUGLYCEMIC HYPERINSULINEMIC CLAMP (EHC). A SUBGROUP OF 50 (OR 26?) obese women….”
-Line 37: “independent correlation factor” these terms cannot be used together. Please check the statistical meaning. I would suggest “the M value was independently and inversely associated with serum Fetuin B”.

-Introduction
Lines 96-99 are misleading. The primary aim of the study was to address the association of serum fetuin B with IR, prediabetes and different metabolic indexes in young women subjected to both 2h-OGTT and EHC. Then, in a subgroup of overweight and obese women, according to reference BMI categories for the Chinese population, the effect of liraglutide 1.8 mg/once daily on fetuin B and glycometabolic homeostasis has been investigated, but this was a secondary aim.
In this respect, I would recommend explaining in depth the rationale behind the choice of liraglutide as the body weight-lowering pharmacological intervention (i.e. why not metformin?). Liraglutide is generally used worldwide at doses up to 1.8mg as antidiabetic agent, but higher doses (up to 3.0 mg) have been approved for weight loss in the US and Europe in obese individuals, irrespective of hyperglycemia. Similar to metformin, off-label use of Liraglutide has been also proposed for the management of PCOS in women with overweight/obesity-related IR. One could wonder if these young overweight/obese women were enrolled for intervention with liraglutide due to ovarian dysfunction and refractory menstrual problems.

-Methods
Lines 110-112: After looking at the metabolic indices in table 1, I believe that the Methods section would benefit of a better definition of the IR-NG study group. Actually, the IR-NG patients were those with insulin resistance at EHC (based on the M-value cut-off < 6.28) but with normal blood glucose levels. I would also suggest stating that the presence of prediabetes in this study was defined by the occurrence of IGT at 2h-OGTT. However, some of the enrolled women had also high fasting blood glucose levels (impaired fasting glucose - IFG).
Line 168: The abbreviation “PLSD” should be described in extenso. Was it a Fisher’s Least Square difference test following up to the analysis of variance?
Lines 170-171: Even though correlation and regression have many similarities, I would recommend using “linear REGRESSION analysis” in this sentence.

-Results
Lines 199-200: I do not feel that the range of serum fetuin B levels in 90.5% of healthy women contributes much.
Lines 204-205: I would suggest using BMI cut-offs that are specific for the Chinese population to define overweight/obesity, and then repeating calculation.
Lines 205-209: This paragraph appear tautological and redundant. It is common knowledge that most patients with prediabetes have IR. I would suggest deleting this paragraph or, at least, merging the IR-NG and IGT patients in single group as a supplementary material, exploring the association of IR with serum fetuin levels by regression analysis (adjusting for blood glucose status) and ROC curve.
Lines 221-224: The model was significant only after controlling for age, probably, due to (known) strong correlations between covariates (age, Fat %, BMI, WC, BP, and TG), that create collinearity problems. In regression analysis, how was multicollinearity handled?
Lines 226: Serum Fetuin B concentrations were divided into four QUARTILES, please revise throughout the manuscript text.
Lines 230-234: I do not feel that the use of row mean score and Cochran-Armitage trend test contributes much, whereas it apparently makes the lecture heavier. Also, “incidence” cannot be used in a cross-sectional context.
Lines 235-239: As anticipated in a previous comment, I would suggest presenting ROC curve analysis as a supplementary material, merging IR-NG and IGT women together. Otherwise, the paradoxical finding of a higher cut-off value of fetuin B in predicting IR-NG that the one predicting IGT (5.44 mg/dL vs 4.45 mg/L) need to be discussed. Indeed, IGT women and not IR-NG women presented the highest concentrations of serum fetuin B in this study.
Lines 247-248: This sentence is tautological, given that IR-NG women were arbitrarily defined by M values below 6.28.
Lines 255: 26 or 50 women? Please check and revise anyplace in the manuscript text/Table 5.

-Discussion
Overall, the discussion reads well. However, I would suggest revising line 279 according to the comments stated above (i.e. redundance of correlation with IR).

-Tables
Please, present the normally distributed data as Means ± Standard Deviations (in spite of standard error).
Table 2: Table title should be revised, given that this is not a Pearson correlation matrix.

-References
Literature review may benefit of two recent references on this topic (adipokines and effects of liraglutide in women, respectively):
- EBioMedicine. 2020;59:102912. doi:10.1016/j.ebiom.2020.102912
- Int J Environ Res Public Health. 2019;17(1):207. doi:10.3390/ijerph17010207

Experimental design

No comment

Validity of the findings

No comment

Reviewer 2 ·

Basic reporting

This article should be correct English.
This manuscript includes sufficient introduction and literature references.

Experimental design

This article got missing several positive controls and validation.

Validity of the findings

Authors should increased serum Fetuin-B levels in women IR and IGT patient groups. However, the totality of the results wasn’t enough to support their conclusions and very limited study.

Additional comments

The manuscript by Xia X. et. al. presents ‘Serum Fetuin-B levels were elevated in young women with 2 pre-diabetes’. This manuscript explored Fetuin-B as a biomarker in pre-diabetic patients. Some studies already reported function of Fetuin-B in metabolic diseases (Meex et al, 2015, Kralisch et al, 2017, and Li et al 2018). Otherwise, controversial papers also suggest (Ebert et al. 2017). In this study, authors investigated the correlation of serum Fetuin-B and human IR and IGT including BMI and lipid metabolism. However, some important data were missing like OGTT and ITT test data. And so, the totality of the results wasn’t enough to support their conclusions and very limited study. There are also some concerns and questions with this manuscript.

Comments
1. In figures, authors didn’t
2. Authors mentioned that most of the previous studies were cross-sectional and descriptive studies. They were merely a preliminary exploration of Fetuin-B on line 293~295. However, they didn’t suggest the reason why previous studies were cross-sectional and descriptive studies. Moreover, they said on line 332~334 like ‘Our research has some limitations, including that 1) our study is a cross-sectional study, so we can not infer the causal relationship between Fetuin-B and the occurrence and development of IR’. These suggestions were not consistent of their hypothesis.
3. They suggest that hyperinsulinemia leads to increased circulating Fetuin-B. However, authors didn’t measure blood insulin levels in this study directly.
4. And authors chose women as the research subjects to avoid the influence of sex hormone on insulin sensitivity in vivo. But generally, women get influence of sex hormone and other hormone compared to men. So, they should confirm those effects on men subjects.
5. In figure 3A, authors measured Fetuin-B levels during OGTT. They suggest that Fetuin-B level is correlated with glucose level and IR level. However, OGTT data was not shown to show the correlation of Fetuin-B and IR or blood glucose level. Moreover, in NGT group, blood glucose level might be decreased after120 min. If Fetuin-B level is correlated with blood glucose level, Fetuin-B level should be decreased to the normal level. But serum Fetuin-B level was not decreased in 120 min time point.
6. In table 1, HbA1c levels in IR-NG and IGT were in normal range below 5.6. So, their IGT and IR-NG patient groups should be validated again.
Minor comments
1. This manuscript should be edited several spelling errors like Fetiun-B on line 322 and grammatic errors.
2. Figure legends were duplicate.

---

## Round 0.2 · Minor Revisions

Although the manuscript has been substantially improved, there are some minor issues that should be addressed, as indicated by reviewer 1.

Reviewer 1 ·

Basic reporting

Manuscript has been considerably improved according to this reviewer suggestion. I have only minor comments:


-Lines 169-171: Please delete this sentence, as the results of Cochran-Armitage trend test and Row Mean Scores analysis have not been included in the revised version.

.To improve readability of this manuscript, I would recommed to report in the introduction section that off-label use of Liraglutide has been proposed for the management of PCOS in women with overweight/obesity-related IR. Also, since women on Liraglutide in this study were affected by ovarian dysfunction and menstrual problems, it would be interesting to mention if there were any clinical improvement in menstrual cycles following pharmacological intervention.

Experimental design

No comment

Validity of the findings

No comment

---

## Round 0.3 · accepted · Accept

The sentence at line 334-335 "And we are recruiting a larger scale of patients in another trial to further study that. " is grammatically incorrect and should be deleted in the galley proofs.